# Efficacy and Safety of Thermal Ablation after Endoscopic Mucosal Resection: A Systematic Review and Network Meta-Analysis

**DOI:** 10.3390/jcm13051298

**Published:** 2024-02-25

**Authors:** Hossein Haghbin, Nuruddinkhodja Zakirkhodjaev, Rawish Fatima, Faisal Kamal, Muhammad Aziz

**Affiliations:** 1Division of Gastroenterology, Ascension Providence Hospital, Southfield, MI 48075, USA; 2Division of Occupational and Environmental Medicine, The University of Texas Health Science Center at Houston, Houston, TX 77021, USA; nurzakir00@gmail.com; 3Division of Rheumatology, University of Toledo, Toledo, OH 43606, USA; rawish.f@gmail.com; 4Division of Gastroenterology, Thomas Jefferson University, Philadelphia, PA 19144, USA; fkamal36@gmail.com; 5Division of Gastroenterology and Hepatology, Bon Secours Mercy Health, Toledo, OH 43608, USA; marajani@hotmail.com

**Keywords:** argon plasma coagulation, colonoscopy, endoscopic mucosal resection, snare tip soft coagulation

## Abstract

(1) **Background**: Large colonic polyps during colonoscopy can be managed by Endoscopic mucosal resection (EMR). To decrease the polyp recurrence rate, thermal ablation methods like argon plasma coagulation (APC) and snare tip soft coagulation (STSC) have been introduced. We performed this network meta-analysis to assess the efficacy and safety of these modalities. (2) **Methods**: We performed a comprehensive literature review, through 5 January 2024, of databases including Embase, PubMed, SciELO, KCI, Cochrane Central, and Web of Science. Using a random effects model, we conducted a frequentist approach network meta-analysis. The risk ratio (RR) with 95% confidence interval (CI) was calculated. Safety and efficacy endpoints including rates of recurrence, bleeding, perforation, and post polypectomy syndrome were compared. (3) **Results**: Our search yielded a total of 13 studies with 2686 patients. Compared to placebo, both APC (RR: 0.33 CI: 0.20–0.54, *p* < 0.01) and STSC (RR: 0.27, CI: 0.21–0.34, *p* < 0.01) showed decreased recurrence rates. On ranking, STSC showed the lowest recurrence rate, followed by APC and placebo. Regarding individual adverse events, there was no statistically significant difference between either of the thermal ablation methods and placebo. (4) **Conclusions**: We demonstrated the efficacy and safety of thermal ablation after EMR for decreasing recurrence of adenoma.

## 1. Introduction

Colonoscopy is the gold standard of colorectal cancer prevention through identification and removal of precancerous lesions [1] Large lateral spreading lesions (LSL) are considered high-risk lesions with higher chances of malignancy over time if left untreated [2]. Endoscopic mucosal resection (EMR) is a safe and effective technique to remove large LSLs, saving the patients from undergoing surgical procedures. Compared to surgery, EMR demonstrates a superior morbidity and mortality profile, in addition to being a better cost-effective procedure [3,4,5]. Despite its superior safety profile, EMR has some adverse events. EMR’s complications include perforation (0.5% of cases) and bleeding, whether during the procedure or in a delayed fashion, with rates of 15.8% and 5.9%, respectively [6,7,8].

Local recurrence of a lesion is a concern after performing EMR that requires follow-up surveillance colonoscopies [9]. This is attributed to multiple factors, most notably adherent macroscopic and microscopic disease not detectable by the endoscopist during the index procedure [10,11]. In support of this, thermal ablation of EMR edges through argon plasma coagulation (APC) or snare tip soft coagulation (STSC) has been shown to reduce the local recurrence rate in subsequent surveillance colonoscopies [12,13]. Thermal energy results in desiccation of cells, resulting in cell death in dysplastic tissue and, hence, less recurrence. STSC uses low-voltage pure sinusoidal current to achieve a more superficial and wider tissue ablation compared to cut current [14]. APC uses the flow of argon gas as a conductor to induce thermal effects in a non-contact manner [15]. For tissue ablation in EMR, APC and STSC generator settings range from 25–65 watts based on location of the lesion in the colon [16]. Through visualization of a 3 mm area of white coagulum after treatment, the endoscopists can identify areas that are adequately ablated and avoid applying energy to exposed muscle fibers to minimize the chance of perforation [16]. At the end of the procedure, endoclips are applied to areas of deep injury to minimize the chance of post-procedure bleeding or perforation [17]. While these methods may decrease recurrence rates, their potential drawbacks include increases in post polypectomy bleeding, post polypectomy syndrome (PPS), and perforation [12].

A recent multicenter three-arm randomized controlled trial (RCT) on 384 patients by Rex et al., comparing APC, STSC, and placebo, found superiority of thermal ablation compared to placebo [16]. Prior systematic reviews and meta-analyses have shown the efficacy and safety of thermal ablation modalities [18,19,20]. However, no updated network meta-analysis incorporating an RCT has been performed to draw comparison between these three modalities. Therefore, we decided to perform this updated network meta-analysis to compare the safety and efficacy of STSC, APC, and placebo.

## 2. Materials and Methods

### 2.1. Search Strategy

This study was conducted per the “Preferred Reporting Items for Systematic Reviews and Network Meta-Analyses (PRISMA-NMA) statement” [21]. We performed an electronic search of the following databases through 5 January 2024: Embase (https://www.embase.com), Medline (PubMed), Web of Science Core Collection (Web of Science, Clarviate), KCI- Korean Journal Index (Web of Science, Clarviate), SciELO, Global Index Medicus, and Cochrane Central Register of Controlled Trials (Cochrane Library, Wiley). The search strategy was defined by H.H. and M.A. This was refined by an experienced medical librarian. The core concepts of “Snare tip soft coagulation”, “Endoscopic Mucosal resection”, “Argon plasma coagulation”, “polyp recurrence”, and "colonoscopy” were used. We provide the search strategy for Embase as an example (Appendix A). The records were exported to EndNote 20 (Clarivate, Philadelphia, PA, USA), and duplicates were eliminated.

### 2.2. Selection Criteria/Included Interventions

All RCTs and prospective/retrospective studies adhering to the following criteria were included: (1) Patients: Adult patients undergoing EMR for colorectal lesions, (2) Outcomes: recurrence rate, adverse events including bleeding, post polypectomy syndrome, and perforation, (3) Comparison/Interventions/Control: STSC, APC, and/or placebo. We excluded reviews, case series, and case reports. We did not restrict the studies based on language or publication date.

### 2.3. Study Screening/Data Extraction

The authors performed the initial screening of studies by titles and abstracts. This screening and subsequent extraction of data were performed by two independent individuals (H.H. and M.A.). In case of any conflicts, the matters were resolved through mutual discussion. Subsequent screening was performed by full text of articles. Population and study characteristics in addition to outcome data were extracted and recorded in Microsoft Excel 2021 (Microsoft, Redmond, WA, USA).

### 2.4. Statistical Analysis

We performed direct and network meta-analyses for the outcomes. Direct meta-analysis was conducted for interventions, with head-to-head results reported in at least two studies. The DerSimonian–Laird approach was the primary tool due to presumed heterogeneity among studies. We used Open Meta Analyst v0.24.1 (CEBM, Brown University, Providence, RI, USA) as the software for direct meta-analysis. The frequentist approach was used for performing the network meta-analysis [22]. For proportional outcomes, the analysis generated the risk ratio (RR), 95% confidence interval (CI), *p*-value and forest plots with placebo as reference. ‘R’ package ‘Netmeta’ version 2.8-2 (Bell labs, Murray Hill, NY, USA) was used for analysis. A random effects model was chosen, given the presumed heterogeneity in studies, and the fixed effects model was used as a sensitivity tool. The intention-to-treat (ITT) protocol was our default assessment for outcomes. The I^2^ statistic was used for heterogeneity, with >50% indicating significant heterogeneity [23]. “Net splitting” was performed to detect differences between direct and indirect evidence for each network outcome (consistency). A *p*-value of < 0.05 was considered statistically significant. The treatment groups were then ranked using P-score, with higher values denoting decreased recurrence and/or adverse events [24].

### 2.5. Bias Assessment

The risk of bias for observational studies and RCTs was performed by risk of bias in non-randomized studies of interventions (ROBINS-I) and the Cochrane risk of bias tool, respectively [25,26]. Quantitative and qualitative publication bias analysis was performed using funnel plots and Egger’s and Thompson–Sharp regression analysis [27,28].

## 3. Results

Using the inclusion/exclusion criteria, a total of 13 studies with 2686 patients were included (Figure 1 and Table 1) [12,13,16,29,30,31,32,33,34,35,36,37,38,39]. Two of the studies were published as a single article [37]. Of these studies, six compared STSC to placebo, four compared APC to placebo, one APC to STSC, and one study compared all three modalities (Table 1). The studies were published between 2002 and 2023. Details and study characteristics are summarized in Table 1.

### 3.1. Direct Meta-Analysis

#### 3.1.1. Snare Tip Soft Coagulation vs. Placebo

Nine studies compared STSC to placebo in terms of lesion recurrence rate. Regarding recurrence rate, STSC showed statistical superiority to placebo (RR: 0.252, CI: 0.199–0.320, *p* < 0.001, I^2^ = 0%) (Appendix A). There was no difference in terms of PPS (two studies, RR: 1.031, CI: 0.108–9.828, *p* = 0.978, I^2^ = 0%), perforation (four studies, RR: 0.566, CI: 0.127–2.528, *p* = 0.456, I^2^ = 9.847%), delayed bleeding (five studies, RR: 0.821, CI: 0.473–1.427, *p* = 0.485, I^2^ = 0%), or intraprocedural bleeding (four studies, RR: 0.573, CI: 0.290–1.135, *p* = 0.110, I^2^ = 84.144%) (Appendix A). Three studies reported total adverse events. Total adverse events in STSC were lower than in placebo (RR: 0.598, CI: 0.439–0.814, *p* < 0.001, I^2^ = 0%) (Appendix A).

#### 3.1.2. Argon Plasma Coagulation vs. Placebo

Five studies compared recurrence rates of APC to placebo. Recurrence rates were lower in APC compared to placebo (RR: 0.350, CI: 0.171–0.715, *p* = 0.004, I^2^ = 11.433%) (Appendix A). There was no difference in terms of PPS (two studies, RR: 0.357, CI: 0.039–3.287, *p* = 0.363, I^2^ = 0%), delayed bleeding (three studies, RR: 0.454, CI: 0.191–1.079, *p* = 0.074, I^2^ = 0%), or intraprocedural bleeding (two studies, RR: 1.421, CI: 0.109–18.500, *p* = 0.789, I^2^ = 31.431%) (Appendix A). Three studies reported total adverse events. Total adverse events of APC did not reach statistical significance compared to placebo (RR: 0.500, CI: 0.250–1.002, *p* = 0.051, I^2^ = 0%) (Appendix A).

#### 3.1.3. Snare Tip Soft Coagulation vs. Argon Plasma Coagulation

Only two studies compared STSC and APC. There were no statistically significant differences between STSC and APC in terms of recurrence rate (RR: 0.837, CI: 0.338–2.069, *p* = 0.699, I^2^ = 37.559%), PPS (RR: 0.760, CI: 0.215–2.690, *p* = 0.670, I^2^ = 0%), perforation (RR: 1.266, CI: 0.326–4.911, *p* = 0.733, I^2^ = 0%), or total adverse event rates (RR: 1.086, CI: 0.588–2.003, *p* = 0.793, I^2^ = 0%) (Appendix A).

### 3.2. Network Meta-Analysis

#### 3.2.1. Recurrence Rate

All studies reported recurrence rates. APC showed good efficacy in reducing the recurrence rate of lesions compared to placebo (RR: 0.33, CI: 0.20–0.54, *p* < 0.01) (Figure 2). Similarly, the use of STSC resulted in reduced recurrence of lesions compared to placebo (RR: 0.27, CI: 0.21–0.34, *p* < 0.01). There was no statistically significant difference between the two interventions (RR: 0.81, CI: 0.48–1.37, *p* = 0.44). On ranking, STSC showed the lowest recurrence rate, followed by APC (Figure 3). The heterogeneity for this analysis was 0%.

#### 3.2.2. Total Adverse Events

Six studies reported the total adverse events. Both APC (RR: 0.53, CI: 0.31–0.91, *p* = 0.02) and STSC (RR: 0.58, CI: 0.43–0.79, *p* < 0.01) demonstrated lower overall adverse events compared to placebo (Figure 4). On ranking, APC showed the lowest total adverse events, followed by STSC (Figure 3). The heterogeneity for this model was 0%.

#### 3.2.3. Post Polypectomy Syndrome

Four studies evaluated PPS. There was no difference between STSC and placebo in terms of PPS (RR: 0.67, CI: 0.10–4.33, *p* = 0.67) (Appendix A). APC was also safe, with no difference compared to placebo (RR: 0.90, CI: 0.13–6.00, *p* = 0.91). On ranking, STSC was ranked the highest, followed by APC and placebo (Figure 3). The heterogeneity for this model was 0%.

#### 3.2.4. Perforation

Five studies reported the perforation rates. Neither APC (RR: 0.71, CI: 0.11–4.71, *p* = 0.72) nor STSC (RR: 0.88, CI: 0.21–3.78, *p* = 0.86) showed higher rates of perforation in comparison to placebo (Appendix A). There was no statistically significant difference between APC and STSC, either (RR: 1.24, CI: 0.33–4.70, *p* = 0.75). On ranking, APC was ranked the highest, followed by STSC and placebo (Figure 3) in terms of lower perforation rates. The heterogeneity for this model was 0%.

#### 3.2.5. Delayed Bleeding

Seven studies reported the delayed bleeding rates. Neither APC (RR: 0.47, CI: 0.20–1.10, *p* = 0.08) nor STSC (RR: 0.82, CI 0.47–1.42, *p* = 0.47) showed higher rates of delayed bleeding in comparison to placebo (Appendix A). On ranking, APC was superior to STSC in terms of delayed bleeding (Figure 3). Thermal ablation interventions were ranked higher than placebo regarding delayed bleeding. The heterogeneity for this model was 0%.

#### 3.2.6. Intraprocedural Bleeding

Seven studies reported the intraprocedural bleeding rates. Neither APC (RR: 0.75, CI 0.17–3.22, *p* = 0.70) nor STSC (RR: 0.61, CI: 0.32–1.17, *p* = 0.13) showed statistically significant different rates of delayed bleeding compared to placebo (Appendix A). On ranking, STSC was ranked the highest, followed by APC and placebo in intraprocedural bleeding (Figure 3). The heterogeneity for this model was 75.6%.

### 3.3. Bias Assessment

Using the ROBINS-I, ranging from low to high, all of the observational studies were noted to have low risk of bias (Appendix A). The Cochran risk of Bias tool showed high risk of bias in blinding of personnel in performance and outcome assessments (Appendix A). This was attributed to lack of blinding of endoscopists during the procedure and in follow-up surveillance. The consistency in primary outcomes of network meta-analysis was evaluated, and no significant difference between direct and indirect evidence was noted (Appendix A). No asymmetry was noted on visualization of funnel plot using recurrence rate as outcome (Appendix A). Further, no significant publication bias was noted using Egger’s regression analysis (*p* = 0.6443) or Thompson–Sharp regression analysis (*p* = 0.7073) (Appendix A). Publication bias for other outcomes was not applicable due to the low number of studies evaluating these outcomes.

## 4. Discussion

This systematic review and network meta-analysis showed the efficacy of thermal ablation in reduced recurrence of lesions after EMR, in addition to the safety of these interventions. Between APC and STSC, there was no statistical significance in terms of safety or efficacy.

Our study reaffirmed the efficacy of thermal ablation in reduced recurrence of lesions. Due to the absence of lymphatics in colonic mucosa, EMR is the preferred modality for most colonic LSLs. Although EMR’s cost-effectiveness and safety are better compared to surgery, EMR’s disadvantage is its higher recurrence rates [40,41]. The recurrence is attributed to macroscopic and microscopic residual tissue; therefore, auxiliary thermal ablation techniques like APC and STSC have been introduced to reduce recurrence rates [10]. In support of this, en bloc resection has significantly lower recurrence rates compared to piecemeal resection, as in the latter, inaccurate sequential snare resections may lead to residual tissue [13]. Emphasizing the need for methods to lower recurrence rates, a large Dutch study showed high recurrence rates of polyps larger than 30 mm when resected piecemeal [42]. US multi-society task force on colorectal cancer grades the use of thermal ablation as a conditional recommendation since prior the literature was only of moderate quality [9]. This study will provide further evidence for future guidelines in favor of recommending thermal ablation.

This network meta-analysis demonstrated the safety profile of thermal ablation after ablation of large non-pedunculated lesions. There was no difference in terms of PPS, perforation, intra-procedural bleeding, or delayed bleeding with thermal ablation compared to placebo. Since APC involves the superficial layers of the gastrointestinal lining, it is considered a safe modality. By the same token, only the snare tip is exposed in STSC. This should limit its ability to cause deep damage. Despite this safety profile, the use of APC with 40 watts of energy and inadvertent contact with tissue have been shown to affect deeper layers including muscularis mucosa, leading to complications like perforation and bleeding [43,44]. There is a need for larger studies and more data regarding the safety profile.

With similar safety and efficacy between APC and STSC, other factors like cost-effectiveness may become more important. Rex et al., in their multicenter RCT, demonstrated that STSC was more cost-effective than APC [16]. In STSC, we use the same snare that was already used for the EMR, whereas APC requires insertion of an APC catheter that costs approximately USD 175 to 275 and requires extra procedure time [16]. We could not assess cost-effectiveness, as the data were not available in other studies. We speculate that the requirement for the APC catheter and extra procedure time, combined with similar efficacy, would translate to inferiority of APC compared to STSC in terms of cost-effectiveness.

Auxiliary EMR techniques other than thermal ablation, such as underwater EMR, have been used to reduce polyp recurrence with an acceptable safety profile. Underwater EMR is a technique in which the lesion is immersed under water, which not only leads to elevation and magnification, but also eliminates the need for submucosal injection prior to resection [45]. In a recent meta-analysis, this method was shown to have less recurrence compared to conventional EMR, but was statistically inferior compared to the STSC group [46]. Of note, underwater EMR results may have been confounded by the use of practices like margin marking before starting the EMR in some of the studies [46]. Margin marking has been shown to lower the risk of recurrence, which is likely due to better delineation of edges and reducing the chance of marginal residual tissue [47]. This is a potential confounder for the efficacy of underwater EMR. In addition, the recent large multicenter trial by Rex et al. was not included in this recent meta-analysis, which could potentially affect the results [16].

Endoscopic submucosal dissection (ESD) is another endoscopic modality that is increasingly used to remove colonic neoplasms. ESD consists of the following steps: marking the margins of the lesion, circumferential incision with needle knife, serial submucosal injection, and hemostasis. ESD removes the neoplasm en bloc, leading to lower rates of recurrence [48]. Despite the aforementioned benefits, drawbacks of ESD include higher perforation rates with surgery requirement, post ESD electrocoagulation syndrome, procedure duration, cost, and limited availability of technical expertise [49,50]. Hence, there is a need for risk stratification before conducting ESD so that the benefits will outweigh the drawbacks. The 2019 American Gastroenterology Association (AGA) Institute clinical practice update encourages consideration of ESD in colonic lesions with features suggesting submucosal invasion like depressed component (Paris 0–IIc) and Kudo type V pit pattern, LSTs larger than two to three centimeters especially in the rectosigmoid area where ESD complications are lower, and last, residual or recurrent adenomas [51]. Therefore, many lesions will still require EMR, emphasizing the need for modalities to reduce recurrence rates. The results of our study are reassuring, as the use of ablation methods can reduce recurrence rates associated with EMR and can expand indications for EMR.

Our study had some limitations. First, due to limitations in the number of studies available, we used both RCTs and observational studies. This will lead to biases inherent to observational studies like confounding bias. However, as it is impossible to blind the endoscopist during the procedures and surveillance in RCTs, the bias gap between observational studies and RCTs may not be as pronounced. This is due to the nature of the intervention, in which endoscopists could not be blinded to the treatment group. This is common to many RCTs in endoscopy. Second, there was heterogeneity in terms of APC modalities, STSC coagulation modes, power settings, and generators of different companies. Despite this, the same ablation mechanism of shallow and wide thermal ablation of tissues with a lower voltage pure sinusoidal waveform holds true for all forms of STSC and superficial ablation for APC modalities [14,15] Another source of heterogeneity is that we pooled data of high volume (>100 patients, *n* = 8) and low volume (<35 patients, *n* = 3) studies that may mirror variable expertise and skills of the endoscopists. There is a need for future studies to evaluate the effect of endoscopic skills in utilization of these modalities, as personal competencies cannot be neglected. Despite heterogeneity being a limitation in meta-analyses, pooling of these diverse data can potentially provide us with a real-world estimate of safety and efficacy not skewed by expertise of a single endoscopist.

## 5. Conclusions

In conclusion, we demonstrated reduced recurrence of lesions and no increase in adverse events with the use of thermal ablation compared to placebo. There was no statistically significant difference between APC and STSC in terms of efficacy or safety. There is a need for more studies to compare these two modalities with other auxiliary techniques and evaluate their cost-effectiveness.

## Figures and Tables

**Figure 1 jcm-13-01298-f001:**
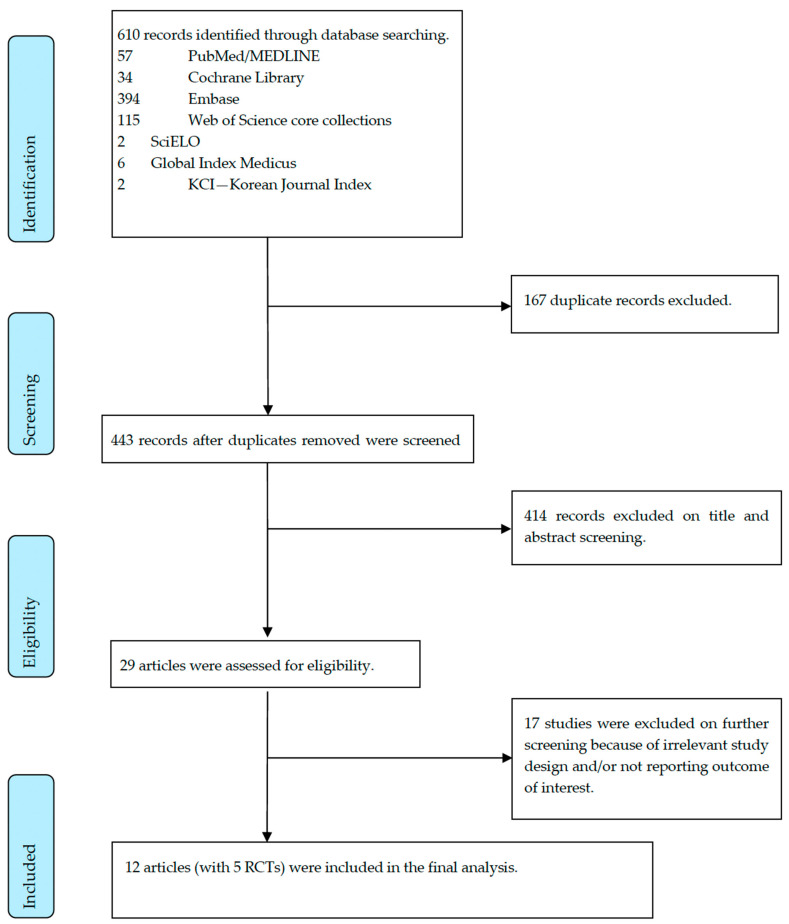
PRISMA flow diagram.

**Figure 2 jcm-13-01298-f002:**
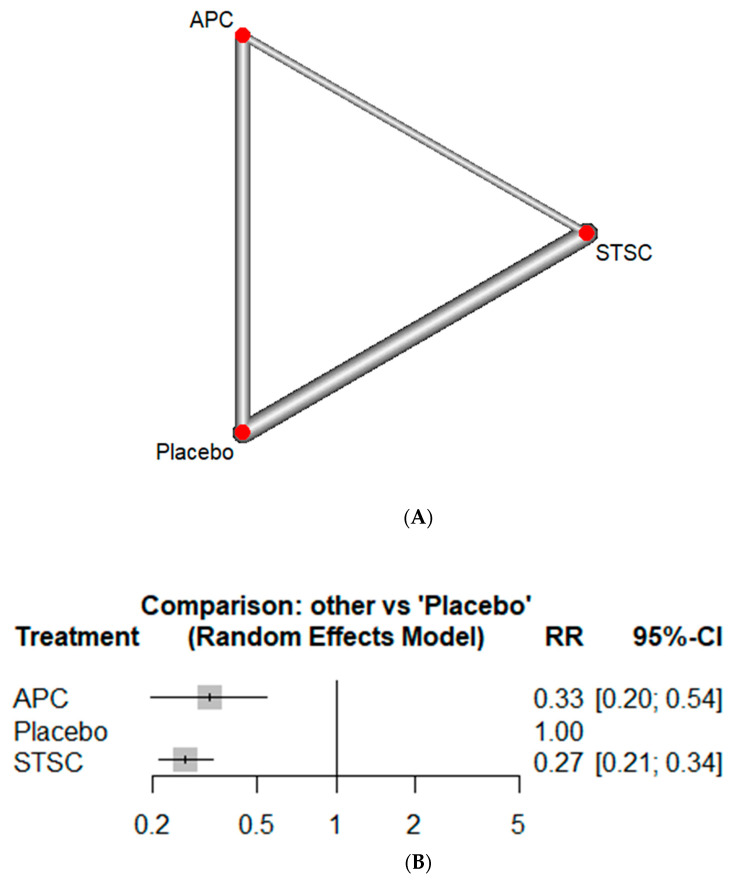
Recurrence rate using network meta-analysis: (**A**) Network diagram (the line represents a direct comparison in studies, and width of line represents number of studies), (**B**) Forest plot with Placebo as comparison group (APC: argon plasma coagulation, CI: confidence interval, RR: relative risk, STSC: snare tip soft coagulation).

**Figure 3 jcm-13-01298-f003:**
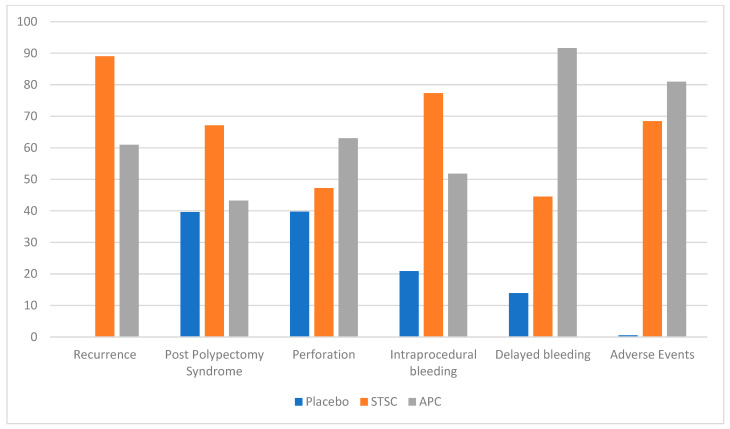
Ranking using frequentist approach and graded using P-score 1–100. Note: Higher P-score represented lower rates of recurrence, post polypectomy syndrome, perforation, intraprocedural bleeding, delayed bleeding, and total adverse events (APC: argon plasma coagulation, CI: confidence interval, RR: relative risk, STSC: snare tip soft coagulation).

**Figure 4 jcm-13-01298-f004:**
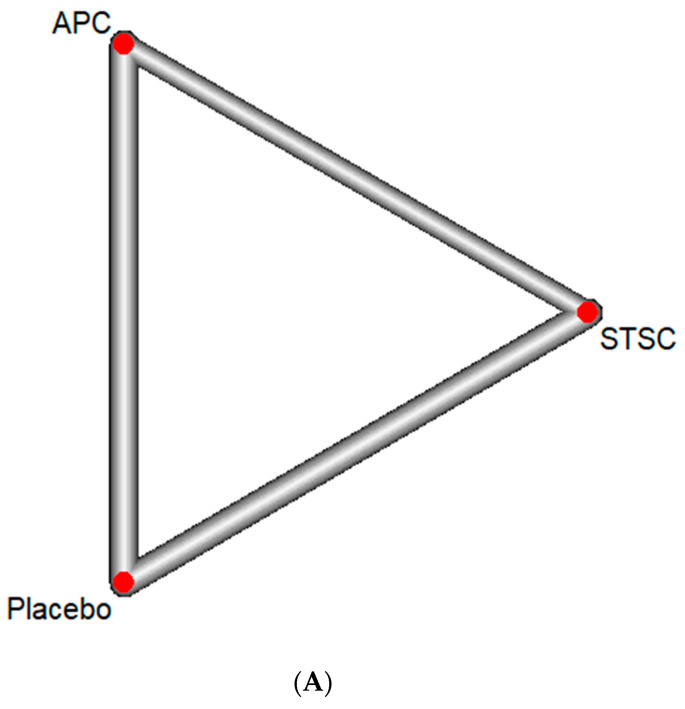
Total adverse events using network meta-analysis: (**A**) Network diagram (the line represents a direct comparison in studies, and width of line represents number of studies), (**B**) Forest plot with Placebo as comparison group (APC: argon plasma coagulation, CI: confidence interval, RR: relative risk, STSC: snare tip soft coagulation).

**Table 1 jcm-13-01298-t001:** Baseline data of finalized studies in this systematic review and network meta-analysis.

Study, Year	Design	Interventions Compared	Morphology/Size	Total Patients	STSC, *n*	APC, *n*	Placebo, *n*
Abu Arisha 2023 [29]	Retrospective and prospective	STSC, Placebo	Non-pedunculated, >2 cm	764	422	NR	342
Albuqurque 2013 [30]	RCT	APC, Placebo	sessile, >2 cm	20	NR	10	10
Brooker 2002 [31]	RCT	APC, Placebo	sessile, >1.5 cm	21	NR	10	11
Groff 2020 [32]	Retrospective	APC, Placebo	NR, >2 cm	53	NR	28	25
Kandel 2019 [33]	Retrospective	STSC, Placebo	NR	120	60	NR	60
Katsinelos 2019 [34]	Retrospective	STSC, APC	LST	101	51	50	NR
Klein 2019 [13]	RCT	STSC, Placebo	LST, >2 cm	416	210	NR	206
Levenick 2022 [12]	Retrospective	APC, Placebo	Non-pedunculated, >2 cm	59	NR	30	29
Nader 2022 [36]	Retrospective	STSC, Placebo	Non-pedunculated, >4 cm	201	133	NR	68
Perez 2021 (1) [37]	Retrospective	STSC, Placebo	>2 cm	76	43	NR	33
Perez 2021 (2) [37]	Retrospective	STSC, Placebo	<2 cm	35	18	NR	17
Rex 2023 [16]	RCT	STSC, APC, Placebo	Non-pedunculated, >1.5 cm	384	126	126	132
Senada 2020 [38]	RCT	STSC, Placebo	LST, >3 cm	148	73	NR	75
Wehbeh 2020 [39]	Retrospective	STSC, Placebo	LST, >2 cm	288	148	NR	140

APC: Argon plasma coagulation, LST: lateral spreading tumor, NR: not reported, RCT: Randomized controlled trial, STSC: snare tip soft coagulation.

## Data Availability

Data are contained within the article and Appendix A.

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
