# Peer review of "Efficacy and Safety of Thermal Ablation after Endoscopic Mucosal Resection: A Systematic Review and Network Meta-Analysis"

_jcm, 2024, doi:10.3390/jcm13051298_

Round 1

Reviewer 1 Report

Comments and Suggestions for Authors

This a well conducted study about the thermal ablation after Endoscopic Mucosal Resection (EMR). The authors made an accurate study of the literature including many studies that were published regarding this subject.

Maybe the introduction could be a little more descriptive regarding the procedure itself and its minuses and possible complications.

Reviewer 2 Report

Comments and Suggestions for Authors

Research regarding efficiency, effect and safety of thermal ablation of resection margins after EMR of colonic lesions. Large-scale systematic review and network-metaanalysis of high expenditure and with clear results.

Different parameters -however- are neglected  and should be mentioned and discussed :

1. Critical analysis of ablation methods:

At least 5 different APC modalities are available on the market with  ( partly completely ) different characteristics, not to speak about different power settings.

Similar factors are relevant for STSC :

Different coagulation modes are present in different rf-generators of different companies. "Soft coagulation"  is not defined uniformly.

2. Different personal competences and skills of endoscopists cannot be neglected

3. Comparison of high-volume studies ( > 100 pts, n=8) vs. low volume studies (<35 pts ,n=3)  is problematic as holds true for mixing RCTs and retrospective studies .

4. ESD as a highly relevant resection method (more and more even in the colon ! ) is not even mentioned not to mention discussed.

Reviewer 3 Report

Comments and Suggestions for Authors

This paper on the efficacy and safety of thermal ablation following Endoscopic Mucosal Resection (EMR) for large colonic polyps is commendable for its thoroughness, robust methodology, and significant contributions to the field. The systematic review and network meta-analysis, encompassing a comprehensive literature review and analysis of 13 studies involving 2686 patients, provide valuable insights into the effectiveness of thermal ablation methods, specifically argon plasma coagulation (APC) and snare tip soft coagulation (STSC). The authors employed a well-established random-effects model and a frequentist approach for the network meta-analysis, ensuring a rigorous statistical evaluation. The findings, which demonstrate a significant reduction in recurrence rates of adenomas with both APC and STSC compared to placebo, are consistent with current literature, adding credibility to the paper's conclusions. The thorough exploration of safety endpoints, including bleeding, perforation, and post-polypectomy syndrome, and the absence of statistically significant differences in adverse events between thermal ablation methods and placebo further strengthen the paper's credibility. Overall, this well-executed study, supported by solid evidence and sound statistical methods, aligns with existing literature and significantly contributes to our understanding of thermal ablation strategies post-EMR. This paper provides valuable insights for clinicians and researchers in the field of endoscopy and colorectal polyp management.

Author Response

The authors would like to thank the reviewer and the editorial team 

Round 2

Reviewer 2 Report

Comments and Suggestions for Authors

All proposals have been adequately incorporated on the revised version.